# A Comprehensive Investigation of Stimulatory Agents on MAIT and Vα7.2+/CD161− T Cell Response and Effects of Immunomodulatory Drugs

**DOI:** 10.3390/ijms25115895

**Published:** 2024-05-28

**Authors:** Parvind Singh, Marianna Száraz-Széles, Sándor Baráth, Zsuzsanna Hevessy

**Affiliations:** Department of Laboratory Medicine, Faculty of Medicine, University of Debrecen, 4032 Debrecen, Hungary; parvind.singh@med.unideb.hu (P.S.);

**Keywords:** MAIT cells, Vα7.2+/CD161− T cells, Cyclosporin A, vitamin D3, stimulation, immunomodulation

## Abstract

Mucosal-associated invariant T (MAIT) cells, a subset of Vα7.2+ T cells, are a crucial link between innate and adaptive immunity, responding to various stimuli through TCR-dependent and independent pathways. We investigated the responses of MAIT cells and Vα7.2+/CD161− T cells to different stimuli and evaluated the effects of Cyclosporin A (CsA) and Vitamin D3 (VitD). Peripheral blood mononuclear cells (PBMCs) from healthy donors were stimulated with various agents (PMA/Ionomycin, 5-OP-RU, 5-OP-RU/IL-12/IL-33) with or without CsA and VitD. Flow cytometric analysis assessed surface markers and intracellular cytokine production. Under steady-state conditions, MAIT cells displayed elevated expression of CCR6 and IL-13. They showed upregulated activation and exhaustion markers after activation, producing IFNγ, TNFα, and TNFα/GzB. CsA significantly inhibited MAIT cell activation and cytokine production. Conversely, Vα7.2+/CD161− T cells exhibited distinct responses, showing negligible responses to 5-OP-RU ligand but increased cytokine production upon PMA stimulation. Our study underscores the distinct nature of MAIT cells compared to Vα7.2+/CD161− T cells, which resemble conventional T cells. CsA emerges as a potent immunosuppressive agent, inhibiting proinflammatory cytokine production in MAIT cells. At the same time, VitD supports MAIT cell activation and IL-13 production, shedding light on potential therapeutic avenues for immune modulation.

## 1. Introduction

MAIT cells, short for mucosal-associated invariant T cells, constitute a subset capable of recognizing microbial-derived riboflavin metabolites present in the major histocompatibility complex (MHC) class I-like molecule MR1. These cells are distinguished by their invariant T cell receptor (TCR) α chain, comprised of a semi-invariant Vα7.2 TCR α chain and a diverse Vβ chain [1]. The presence of this semi-invariant Vα7.2 TCR α chain enables MAIT cells to identify riboflavin metabolites, such as 5-riboflavin phosphate and 5-formyl riboflavin, produced by microorganisms and displayed on MR1 molecules. Predominantly localized in the liver, spleen, and gastrointestinal tract [2], MAIT cells also constitute up to 10% of T cells in peripheral blood [3], with no significant gender-based differences observed [4]. These cells are pivotal in the innate immune response against microbial infections. MAIT cells exhibit responsiveness to various activation methods in vitro. TCR-dependent activation involves stimulation with microbead- or plate-bound anti-CD3/anti-CD28 antibodies, 5-(2-oxopropylideneamino)-6-d-ribitylaminouracil (5-OP-RU), or paraformaldehyde-fixed *E. coli* in the presence of antigen-presenting cells (APCs). TCR-independent activation includes direct stimulation with recombinant cytokines IL-12 and IL-18, and indirect stimulation with TLR-4/TLR-8 agonists or influenza A virus in the presence of APCs [5,6,7]. Currently, human MAIT cells are identified using 5-OP-RU loaded MR-1 tetramers or by the co-expression of CD161++ and TCR Vα7.2-Jα33+ [8]. In the Vα7.2+ T cell compartment, a subset, denoted as CD161-, exists. The functional role of Vα7.2+/CD161− T cells is unclear, but studies have shown an increase in chronic HCV patients compared to healthy controls, while MAIT cells were compromised [9]. Similar findings have been reported in chronic HIV-1-infected patients, where Vα7.2+/CD161− cells accumulated while MAIT cell frequency decreased compared to uninfected healthy controls, despite no change in total T cell numbers [10]. Recently, we have described them in atopic dermatitis cases, where they have been found as Th22-producing cells with higher CD38 and lower IFNγ-producing T cells, and they are closer to conventional T cells despite expressing the Vα7.2 marker [11].

TCR-dependent activation of MAIT cells prompts the release of cytotoxic substances, such as perforin and granzyme B, alongside the secretion of pro-inflammatory cytokines and chemokines. Conversely, TCR-independent activation is instigated by cytokines like IL-12 and IL-18, predominantly resulting in the expression of IFN-γ, perforin, and granzyme B under the regulation of promyelocytic leukemia zinc finger (PLZF) and T-bet [12,13]. Notably, TCR-dependent activation instigates a tissue-repair program, whereas TCR-independent activation tends to yield a Th1-like phenotype [6,7]. MAIT cells exhibit responsiveness to various stimuli, including Phorbol 12-myristate 13-acetate (PMA)/Ionomycin, 5-OP-RU, and IL-12/IL-33, among others, culminating in cytokine production and T cell activation through diverse pathways. The specific response of MAIT cells to these stimuli may vary, with some promoting pro-inflammatory reactions and others potentially influencing MAIT cell differentiation and activation [14,15,16]. IL-33 works as an alarmin in inflammation and tissue homeostasis. Recently, it has been described that IL-33 alone can upregulate CD69, and together with IL-12, they upregulate TNFα and IFNγ-producing MAIT cells, leading to the pro Th1 effect [16].

Immunomodulatory drugs such as lenalidomide and pomalidomide have been identified as indirect inhibitors of MAIT cell activation [17]. A few studies found that MAIT cells may respond to immune checkpoint blockade (ICB) in vitro. MAIT cells from MM patients express high PD-1 and show increased functionality upon PD-1 blocking in vitro; similar results were documented for prostate cancer patients [18,19]. A recent report suggested that MAIT cells may respond to PD-1 blockade in vivo in a small cohort of metastatic melanoma patients [20]. This inhibition of MAIT cell activation by immunomodulatory drugs underscores the intricate interplay between these agents and the immune system, implying a possible influence on the therapeutic efficacy of MAIT cells.

Cyclosporin A (CsA) and Vitamin D3 (Vit D) exert distinct effects on T cells in vitro. CsA, an immunosuppressive drug, inhibits calcineurin, thereby suppressing the nuclear factor of activated T cells (NF-AT) and subsequently reducing the production of cytokines such as IL-2, IL-4, interferon-gamma, and TNF-alpha [21,22]. This immunosuppressive action of Cyclosporin A impacts T-cell activation and proliferation by blocking crucial signaling pathways essential for immune responses [23]. Conversely, Vitamin D3 has been observed to significantly diminish CD4+ T-cell activation compared to low-dose vitamin D3, as indicated by reduced intracellular CD4+ ATP release in a randomized controlled trial. This decline in CD4+ T-cell activation suggests that Vitamin D3 may influence cell-mediated immunity, although specific effects on T cells and their functions are not explicitly known [24]. The inhibition of NF-AT by Cyclosporin A could affect MAIT cell function by impeding their activation and ability to produce cytokines vital for immune responses. Vitamin D3 influences various immune cells by modulating their differentiation, maturation, and function. Studies have shown the suppressive effects of Vitamin D3 on NF-kB signaling pathways in T cells, monocytes, and macrophages, altering T cells’ proliferation and differentiation profile. Vitamin D3 fosters an anti-inflammatory milieu by inhibiting pro-inflammatory Th1 and Th17 cells while promoting the differentiation of Th2 and Treg cells [25]. Additionally, studies have linked MAIT cell frequencies with serum Vitamin D3 concentrations, suggesting a potential correlation between Vitamin D3 levels and MAIT cell activity in conditions such as asthma [26].

In our study, we have investigated the activation and cytokine production of MAIT cells and Vα7.2+/CD161− T cells in response to the various stimulatory conditions, including PMA/Ionomycin, 5-OP-RU, and 5-OP-RU/IL-12/IL-33. Furthermore, we showed the effects of immunomodulatory drugs (cyclosporin A and Vitamin D3) on MAIT cells and Vα7.2+/CD161− T cells based on different stimulatory conditions.

## 2. Results

### 2.1. Effects of Different Stimulants on MAIT and Vα7.2+/CD161− T Cells

To compare the effects of different stimulants, we used non-treated cells (without CsA and VitD treatment) for calculating the statistical significance to avoid treatment biases within the data. MAIT cells are CCR6-expressing cells under steady-state conditions. However, a significant decrease was noted under different stimulatory conditions, specifically with PMA/Ionomycin. Negligible CCR6 expression was found in Vα7.2+/CD161− T cells and no changes were observed in response to stimulants. CD69 is an acute activation marker, and as expected, sharp increases were noted in the case of every stimulant compared to unstimulated MAIT cells; the highest expression was noted in 5-OP-RU and 5-OP-RU along with cytokines (IL-12 and IL-33). In Vα7.2+/CD161− T cells, only PMA stimulation increased CD69 expression, but CD69 expression did not show significant changes in response to other stimuli compared to unstimulated cells. CD38 is a chronic activation marker, and MAIT cells showed higher CD38 expression with different stimulants than unstimulated cells, whereas Vα7.2+/CD161− T cells showed no significant differences. Interestingly, PD-1/CD69 expression of MAIT cells was significantly increased by 5-OP-RU and 5-OP-RU/IL-12/IL-33 stimulation, whereas the proportion of double-positive (PD-1/CD69) Vα7.2+/CD161− T cells was significantly increased by only PMA (Figure 1).

A slight increase in PD-1 expression was visible in MAIT cells. However, the amount of elevation was not substantial, and no significant results were noted for Vα7.2+/CD161− T cells. MAIT cells were consecutively decreased post-activation, but the extent of the decrease was insignificant. However, the Vα7.2+/CD161− T cell subpopulation did not show a reduction. We previously showed that PB MAIT cells did not express CLA and found similar results after different stimulatory conditions. However, PMA-stimulated Vα7.2+/CD161− T cells enhanced the CLA expression on the surface compared to other stimulants. Interestingly, both CLA and CCR6 expression on the surface of Vα7.2+/CD161− T cells increased following PMA stimulation, whereas this was not observed in MAIT cells (Appendix A).

### 2.2. Cytokine Production Fluctuates with Varied Stimuli

In steady-state conditions, MAIT cells are Th2-type cells expressing more IL-13 and no IFNγ. However, a significant decrease in IL-13 and an increase in IFNγ post-stimulation are noted. Whereas Vα7.2+/CD161− T cells were not producing any type of cytokines in a steady-state condition, as expected, significant increases in IFNγ are found in the case of PMA stimulation. Polyfunctional MAIT cells (IFNγ/IL-13) were absent in steady-state conditions. However, MAIT cell ligand interaction produced a significantly higher frequency of these polyfunctional MAIT cells, which increased more in the case of proinflammatory cytokines and the MAIT cell ligand stimulus (5-OP-RU/IL-12/IL-33). The proportion of IL-17A-producing MAIT cells increased slightly, but the data were not statistically significant, whereas PMA-stimulated Vα7.2+/CD161− T cells showed a significant difference. 5-OP-RU/IL-12/IL-33-stimulated TNFα-producing MAIT cells showed no significant difference compared to unstimulated controls, while significant differences were observed in PMA and 5-OP-RU-stimulated cells. The production of TNFα by Vα7.2+/CD161− did not change T cells compared to controls. Interestingly, stimulation with 5-OP-RU with or without IL-12/IL-33 resulted in a significantly higher amount of polyfunctional TNFα/GzB-producing MAIT cells, whereas no significant difference was noted in Vα7.2+/CD161− T cells (Figure 2).

Only PMA-stimulated MAIT cells and Vα7.2+/CD161− T cells produced significantly higher IL-22; the rest of the stimulants could not reach a statistically significant difference. Similar results were found for the IL-17A+/IL-22+ subpopulation of MAIT cells and Vα7.2+/CD161− T cells, where statistically significant upregulation was only observed for PMA. Since, in the case of Vα7.2+/CD161− T cells, they were only stimulated in the case of PMA/Ionomycin, taken together, IL-17A-, IL-22-, and IL-17A/IL-22-producing Vα7.2+/CD161− T cells only showed a difference with PMA and not with ligand (5-OP-RU). No significant differences were noted in Granzyme B (GzB)-producing MAIT cells and Vα7.2+/CD161− T cells with stimulants. We could not detect IL-4-producing MAIT cells; however, Vα7.2+/CD161− T cells expressed significantly higher amounts of IL-4 and IFNγ/IL-4 post-PMA stimulation (Appendix A).

### 2.3. Cyclosporin A and Vitamin D3 Modulatory Changes in the Cellular Surface of MAIT and Vα7.2+/CD161− T Cells under Different Stimulation

No significant effect of immunomodulatory drugs was noted within MAIT cell frequency. However, 5-OP-RU-stimulated and CsA-treated Vα7.2+/CD161− T cells showed significantly higher frequencies than the untreated cells. CsA significantly downregulated the CCR6 expression of MAIT cells, specifically when they were stimulated with the 5-OP-RU ligand in the presence of proinflammatory cytokines (IL-12 and IL-33), and no significant differences were noted in Vα7.2+/CD161− T cells. VitD treatment upregulated the CD69 expression of MAIT cells when stimulated with PMA, while with 5-OP-RU stimulation, CsA significantly upregulated the CD69 expression. No difference was found in CD69 expression of Vα7.2+/CD161− T cells compared to untreated control. 5-OP-RU ligand-specific changes were noted in CD38 expression; the opposite effect of the stimulants was detected, whereas CsA showed downregulated and VitD showed upregulated expression of CD38 in MAIT cells. There was no difference in CD38 expression by Vα7.2+/CD161− T cells during treatment. CsA and VitD modified PD-1/CD69 expression in MAIT cells in the presence of IL-12 and IL-33. The proportion of PD-1/CD69-positive MAIT cells was lower in the presence of CsA and VitD, while no difference was detected in Vα7.2+/CD161− T cells (Figure 3).

PD-1 expression of MAIT cells remained stable with immunomodulatory drug treatment. However, VitD treatment showed significantly reduced PD-1 expression during PMA stimulation on Vα7.2+/CD161− T cells. No differences in CLA expression were noted in MAIT and Vα7.2+/CD161− T cells upon CsA and VitD treatments. CsA modulated and downregulated the CLA/CCR6 expression of MAIT cells in the presence of cytokines (IL-12 and IL-33), and no effects of drugs were found in Vα7.2+/CD161− T cells (Appendix A).

### 2.4. Cyclosporin A and Vitamin D3 Modulatory Changes in Cytokine Production of MAIT and Vα7.2+/CD161− T Cells under Varied Stimulation

The production of IFNγ by MAIT cells was induced by PMA, 5-OP-RU, and 5-OP-RU combined with the cytokines IL-12 and IL-33 in different amounts. CsA blocked IFNγ production of MAIT cells in every case. However, VitD has a similar effect only in the case of 5-OP-RU. As expected, PMA-stimulated Vα7.2+/CD161− T cells also showed IFNγ blockade with CsA treatment. VitD promoted the IL-13 production by MAIT cells when cells were stimulated in a 5-OP-RU-dependent manner, whereas no significant treatment differences were noted in IL-13 production of Vα7.2+/CD161− T cells. The sharp blocking was noted in polyfunctional IL-13/IFNγ-producing MAIT cells during ligand-specific stimulation in the case of CsA and VitD treatment. A similar observation was pointed out in the PMA stimulation of Vα7.2+/CD161− T cells. IL-17A/IL-22+ MAIT cells did not show any treatment-related changes. However, VitD-treated, unstimulated Vα7.2+/CD161− T cells showed increased polyfunctional IL-17A/IL-22-producing cells. A CsA-dependent decline in TNFα-producing MAIT cells and Vα7.2+/CD161− T cells was noted with PMA-specific stimulation; CsA could not help with MAIT cell-specific stimulants. Inversely, GzB/TNFα-producing MAIT cells were blocked by CsA and VitD when stimulated with 5-OP-RU. Interestingly, no drug effect was noted in the 5-OP-RU ligand with IL-12/IL-33 stimulation. Treatment-based inhibition was noted in GzB/TNFα-producing Vα7.2+/CD161− T cells (Figure 4).

No significant changes in immunomodulation were noted with IL-17A-producing MAIT cells; interestingly, CsA-treated, IL-17A-producing Vα7.2+/CD161− T cells were increased when stimulated with the 5-OP-RU ligand. No treatment-based significant differences were noted in IL-22-producing MAIT and Vα7.2+/CD161− T cells. MAIT cells did not express IL-4, and no difference based on treatment was found in IL-4- and IFNγ/IL-4-producing Vα7.2+/CD161− T cells. The treatments did not affect GzB-producing MAIT cells and Vα7.2+/CD161− T cells (Appendix A).

## 3. Discussion

MAIT cells are known for their vast immune response; they exhibit distinct polarization profiles when isolated from different barriers, suggesting varied functions in blood and mucosal barriers. The responses of MAIT cells can be influenced by the stimulation conditions and the tissue microenvironment they encounter [27]. Recently, Konecny et al. summarized MAIT cell signals that integrate different functions in healthy and inflamed tissues. They also highlighted the complexity of the computational exploration of cell–cell communication using single-cell data [28]. Fergusson et al. reported that MAIT cells are characterized by high expression levels of multi-drug resistance protein 1 (MDR1), an ATP-binding cassette-multi-drug efflux protein. This high MDR1 expression shown by MAIT cells confers resistance to cytotoxic substances like daunorubicin, protecting from daunorubicin-induced apoptosis. However, MAIT cells are not shielded from the antiproliferative and cytotoxic effects of immunosuppressive substances [29]. CD161 is a C-type lectin-like receptor (CD161) used as a marker to identify MAIT cells, along with the Vα7.2 TCRα chain [30]. The Vα7.2+/CD161− T cell subset represents a distinct population that differs from MAIT cells. These cells have been reported to have different gene expression patterns compared to MAIT cells despite sharing the Vα7.2+ compartment [31]. We recently reported these cells to have different responses in atopic dermatitis cases than the MAIT cells, where Vα7.2+/CD161− T cells were showing a classical Th2/22 response and MAIT cells were showing upregulation in polyfunctional TNFα/GzB-producing cells [11]. We have investigated activation, migration, exhaustion, and several cytokines under commonly used stimulatory conditions known to MAIT cells. The effects of widely used immunomodulatory drugs during these stimulatory conditions are essential to understanding MAIT cell biology. We also compared the Vα7.2+ T cell compartment, which is CD161-, to see how different they are from MAIT cells (CD161+).

Post-stimulation, a nonsignificant decrease in MAIT cell frequency was observed, which might be explained by TCR internalization from the immunological synapse, which is necessary for sustained TCR signaling and TCR downregulation, which enhances T cell proliferation in response to stimulation [32]. We have summarized the significantly different results based on various treatments for given stimulants in Figure 5. Our results are in coherence with previously published data that showed high CCR6 chemokine expression by MAIT cells [2]; however, a significant reduction was observed post-stimulation and CsA treatment inhibited the expression in the presence of 5-OP-RU/IL-12/33. Juno et al. showed a loss of CXCR3 expression, which was compensated by the increased expression of CCR6 and CXCR6 in renal patients [33]. To extrapolate that there is the possibility of CCR6 compensation with some other migratory chemokines, it would have been interesting to investigate known MAIT cell migratory chemokine receptors such as CCR9, CCR5, and CXCR6 under the same conditions [27,33]. 5-OP-RU with or without IL-12/IL-33 upregulates the CD69 and CD38, and a CsA-dependent increase in CD69 expression was noted in the presence of 5-OP-RU. CD69 is a marker of early T cell activation, while CD38 is associated with T cell activation and proliferation [34,35]. Interestingly, CsA showed a decrease and VitD showed an increase in CD38 expression. Pincikova et al. reported randomized VitD supplementation in 16 cystic fibrosis patients. They evaluated the MAIT cells, which correlated positively between VitD treatment and CD38 expression, and a negative correlation with PD-1-expressing MAIT cells was found [36]. These data support our finding that VitD has robust immunomodulatory effects and can be translated to clinical conditions. PD-1/CD69 cells are highly upregulated in 5-OP-RU with or without IL-12/IL-33. CsA and VitD have a high level of control over CD69/PD-1 expression in the IL-12/IL-33-stimulated cells.

TCR-dependent activation of MAIT cells alone in the absence of proinflammatory cytokines could not maintain sustainable IFNγ production, leading to the upregulation of tissue-repair-associated genes [37,38]. We found that IFNγ-producing MAIT cells were severely blocked by the presence of CsA and VitD with PMA and 5-OP-RU. Interestingly, only CsA could block IFNγ in the presence of proinflammatory cytokines and TCR stimulation. CsA blocks TCR-mediated calcium signaling pathways and has been shown to block IFNγ-producing MAIT cells [39]. IL-13-producing MAIT cells are downregulated as an effect of stimulation and are severely affected in the presence of IL-12/IL-33. Interestingly, polyfunctional MAIT cells producing IL-13/IFNγ are significantly boosted by proinflammatory cytokines. Kelly et al. found chronically (Phytohaemagglutinin, IL-2, IL-7, anti-CD3/28, and later PMA/Ionomycin) stimulated MAIT cells showing IL-13 expression. However, in flow cytometric analysis, unstimulated MAIT cell control was missing; instead, conventional T cell control was used as the culture control. Interestingly, GATA-3 expression (an IL-13-associated gene) showed no significant difference between before and after stimulation of MAIT cells [40].

The plasticity inherent in MAIT cells, rather than being strictly lineage-derived, enables them to dynamically assimilate signals from their environment, perpetually adapting and altering their functions. This transcriptional plasticity depends on factors such as tissue localization, clonal identity, and activation status [41,42,43]. VitD increases the IL-13-producing MAIT cells in a 5-OP-RU-dependent manner, and it was shown by Hinks et al. that MAIT cells stimulated through TCR upregulated the Vitamin D receptor in an in vitro model and mice, too [13]. These data support Vit-D’s immune balance by blocking the IFNγ and upregulating the IL-13 from MAIT cells to obtain a steady-state condition while being stimulated by the MR-1 ligand, without proinflammatory cytokines. In peripheral blood, a minor population of MAIT cells produces IL-17A; however, in tissues such as the liver and female reproductive tract, large numbers of IL-17A- and IL-22-producing MAIT cells are detected upon *E. coli* stimulation [44,45]. Buccal mucosa MAIT cells also produce more IL-17 than the blood MAIT cells. Similarly, PMA-stimulated decidual MAIT cells produce very limited IL-17 and almost no IL-22 in comparison to female genital mucosal MAIT cells [46,47]. Our data are in coherence with those of the previously mentioned studies. We found that PB MAIT cells produce very limited IL17A, IL22, and polyfunctional IL17A/IL-22 post-PMA stimulation.

Flow cytometric MAIT cell detection is performed with Vα7.2 expression with high expression of c-type lectin CD161. The discovery of the MR-1 ligand enabled the development of a 5-OP-RU-loaded MR-1 tetramer, which is used to detect TCR-specific MAIT cells called MR-1 restricted cells. Not all MAIT cells identified with Vα7.2/CD161+ bind to the MR-1 tetramer; not all MR-1 binding tetramer express Vα7.2/CD161+ [48]. Recently, another host-derived MR-1 ligand was identified called cholic acid 7-sulfate (CA7S), recognized by MAIT cells. In contrast to 5-OP-RU, CA7S promoted MAIT cell survival and an upregulated gene associated with wound healing and immunoregulation [49]. In addition to our previous results [11], this in vitro study compares MAIT cells (Vα7.2+/CD161+) and Vα7.2+/CD161− T cells under different stimulatory conditions and treatments. The results revealed that these two populations differ despite sharing a Vα7.2 compartment. However, since this population was only stimulated with the 5-OP-RU ligand and not with the newly reported CA7S ligand, it would be interesting to examine stimulation with both ligands together on MAIT and Vα7.2+/CD161− T cells, which would provide more details about these populations. Park et al. showed, with RNA sequencing of TCR Vα7.2+CD161− T cells, that they are markedly different from Vα7.2+/CD161+ cells and more similar to the conventional T cells. However, MAIT and Vα7.2+/CD161− T cells share some common upregulated and downregulated genes [50]. Kathrin et al. showed the CDR3 repertoire of Vα7.2+ chains and found dominating clonal expansion of canonical and noncanonical clones within MAIT cells, whereas Vα7.2+/CD161− subsets were polyclonal [51]. Our results support the previous evidence that different types of MAIT cell activation result in different responses. Proinflammatory cytokine-supported 5-OP-RU ligand-based MAIT cell activation produced the highest CD69 expression and Th1 type response producing IFNγ. A polyfunctional response of MAIT cells was also observed with the same stimulation, such as the highest PD-1/CD69, IL-13/IFNγ, and TNFα/GzB. Cyclosporin A is a potent immunosuppressing agent that significantly reduces the migration and activation of several cytokines in the presence of different MAIT cell stimulations. Vα7.2+/CD161− T cells were not activated with the 5-OP-RU ligand with or without IL-12 and IL-33. With PMA stimulation, they produced higher amounts of IL-17A, IL-4, IL-17A/IL22, and IFNγ/IL-4, which the MAIT cells produce in negligible numbers.

We comprehensively investigated the MAIT cell’s response to CsA and Vit D upon different stimulations, which is essential to understanding the immunomodulatory effect under different stimulatory conditions. However, our study did not include a recently identified CA7S ligand, which is known to enhance survival and upregulate genes associated with wound healing and immunoregulation.

In conclusion, MAIT cells are a different population than Vα7.2+/CD161− T cells, which are closer to conventional T cells. Cyclosporin A is a potent immunosuppressing agent that significantly blocks the proinflammatory cytokines in MAIT cells, while vitamin D supports MAIT cell activation and IL-13 production.

## 4. Materials and Methods

### 4.1. Sample Collection and Peripheral Blood Mononuclear Cells (PBMCs) Isolation

First, 18 mL of peripheral blood (PB) was collected through venipuncture in a BD vacutainer containing sodium heparin anticoagulant (Becton Dickinson, San Jose, CA, USA) from 9 healthy control donors.

### 4.2. PBMC Isolation

PBMC was isolated from PB collected in a heparin tube using the density gradient centrifugation method. Briefly, 3 mL of Ficoll–Hypaque (Merck KGaA, Darmstadt, Germany) was added to a 15 mL falcon tube, and heparinized blood was layered upon it gently. The tubes were centrifuged at 1500 RPM for 30 min at room temperature. Post-centrifugation, only a buffy layer was isolated in another tube and dissolved in RPMI-1640 media containing 1% streptomycin/penicillin and 10% fetal bovine serum (Merck KGaA, Darmstadt, Germany). Cells were rewashed with centrifugation at 1500 RPM for 5 min before performing cell count and were adjusted to around 1 × 10^6^–2 × 10^6^ cells/100 μL.

### 4.3. Reagent Preparation

5-OP-RU preparation: 5-OP-RU are potent activators of MAIT cells and are formed by non-enzymatic reactions between 5-amino-6-d-ribitylaminouracil (5-A-RU), an early intermediate in bacterial riboflavin synthesis, and glyoxal or methylglyoxal. We prepared 5-OP-RU by mixing a 1:50 ratio of 5-A-RU and methylglyoxal. 5-A-RU was purchased commercially from Cayman Chemical (Ann Arbor, Michigan, USA), and the Methylglyoxal solution (40% H_2_O) was purchased from Merck (KgaA, Darmstadt, Germany). 5-OP-RU were made fresh before the experiment and diluted in Dimethyl sulfoxide (DMSO) (Merck KGaA, Darmstadt, Germany) where 5-A-RU and methylglyoxal were incubated for 2 h at room temperature. Moreover, 10 nM/mL of the final 5-OP-RU concentration stimulated MAIT cells.

Cytokines: IL-12 and IL-33 cytokines were purchased from Biolegend (San Diego, CA, USA), diluted in RPMI media, and used in the experiment at a final concentration of 50 ng/mL.

Phorbol 12-myristate 13-acetate (PMA)/Ionomycin: PMA was diluted in RPMI media, and a final 20 ng/mL concentration was used. Ionomycin was used at 1.5 μg/mL. Furthermore, 10 ng/mL of Brefeldin-A was used diluted in RPMI.

Immunomodulatory drugs: Vitamin D3 and Cyclosporin A were purchased from Merck (KgaA, Darmstadt, Germany). Vitamin D3 was diluted in ethanol; the final concentration was 0.1 nM/mL. Cyclosporin A was diluted in RPMI media, and the final concentration was 400 ng/mL.

### 4.4. Experiment Design

The experiment was performed in a FACS tube containing 1 × 10^6^–2 × 10^6^ PBMCs/100 μL, and the total volume was adjusted to 1 mL with RPMI. The complete experiment design and setup are described in Table 1. All the tubes were vortexed and incubated at 37 °C with 5% CO_2_ for 18 h, followed by Brefeldin A being added, and incubation continued for six more hours. Finally, cells were washed, and cellular surface and intracellular staining were performed using monoclonal antibodies.

### 4.5. Flow Cytometry

Three panels of monoclonal antibodies were used to perform cellular surface and intracellular staining (Table 2). Surface staining was performed by adding monoclonal antibodies in an appropriate concentration to PBMCs, which were incubated for 15 min at an ambient temperature in the dark and then washed with phosphate-buffered saline (PBS).

Intracellular tubes were added 100 μL of Intraprep permeabilization buffer 1 (Beckman Coulter Brea, CA, USA), vortexed, and incubated again for 15 min. Tubes were washed with 4 mL of PBS using centrifugation (1500 RPM for 5 min) and the PBS was decanted. Then, 100 μL of Intraprep permeabilization buffer 2 was added and incubated for 5 min (without vortexing). Then, fluorescently labeled cytoplasmic antibodies were added and incubated for 30 min in the dark at room temperature. After incubation, 4 mL of PBS was added to cells, washed with centrifugation, and diluted with 400 μL of paraformaldehyde for acquisition.

All the acquisitions were performed on the BD FACS Canto II (Franklin Lakes, NJ, USA) flow cytometer. Daily internal quality control checkup was performed using a cytometer setup and tracking beads (CS&T) measured to keep equipment performance tracking. External quality control assessment of equipment and process was performed by participating in the UK-NEQAS Leukemia immunophenotyping program. Tubes were acquired using a carousel setting or manually with medium to high acquisition speed, and a stop gate was set for 200,000 events for each tube.

### 4.6. Data Analysis

Flow cytometric data were analyzed using sequential gating of MAIT cells and Vα7.2+/CD161− T cells (Appendix A). Surface and intracellular expression were gated within MAIT and Vα7.2+/CD161− T cells using FlowJo v10.9.0 software (Appendix A). The % of MAIT and Vα7.2+/CD161− T cells among T cells and the % of CD69, PD-1, CD69/PD-1, CCR6, CLA, CCR6/CLA, IFNγ, IL-4, IL-13, IFNγ/IL-4, IFNγ/IL-13, IL-17, IL-22, IL-17/IL-22, CD38, GzB, TNFα, and GzB/TNFα among MAIT and Vα7.2+/CD161− T cells were exported into a spreadsheet and the data were statistically analyzed.

### 4.7. Statistical Analysis

Statistical analysis was performed using GraphPad Prism statistical software 9.0 (GraphPad Software, San Diego, USA). The Kolmogorov–Smirnov (SK) test was used to test the normality of the data distribution. A two-way ANOVA was performed to compare different stimulations and treatments. The statistical significance of the findings was set at a *p*-value of less than 0.05.

## Figures and Tables

**Figure 1 ijms-25-05895-f001:**
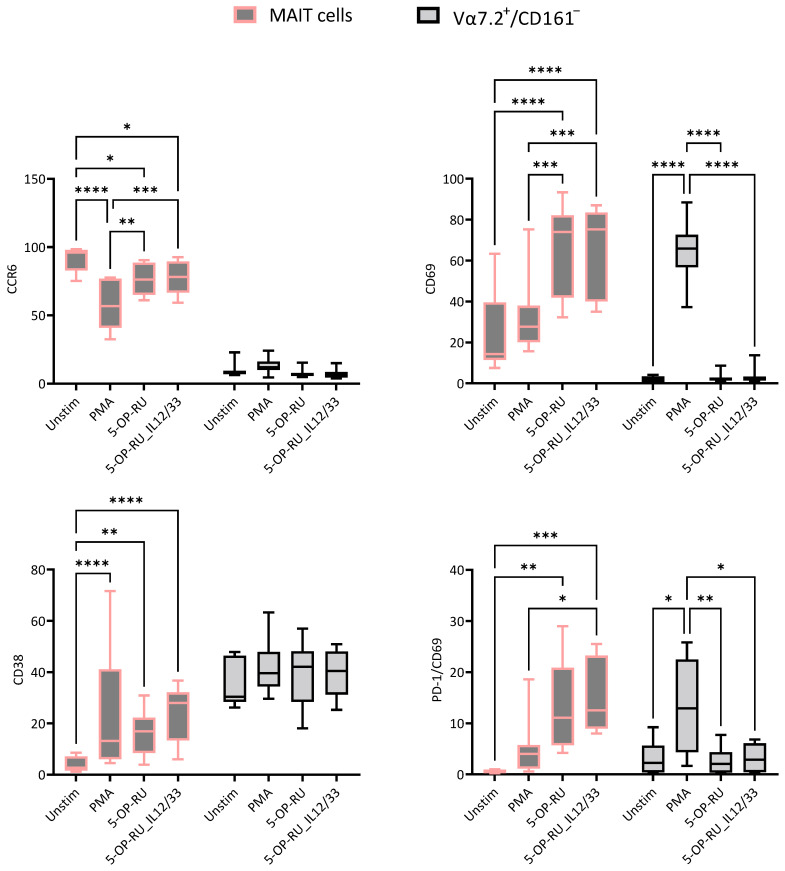
Cellular surface expression of MAIT and Vα7.2+/CD161− T cells was compared between different stimulatory conditions. Grey color gradients with pink borders are MAIT cells, and grey color gradients with black color borders represent Vα7.2+/CD161− T cells. X-axis represents different stimulatory conditions (Unstimulated, Phorbol myristate acetate (PMA)/ionomycin, 5-(2-oxopropylideneamino)-6-d-ribitylaminouracil (5-OP-RU), and 5-OP-RU/IL-12/IL-33) and Y-axis represents marker expression within as % of MAIT cells or % of Vα7.2+/CD161− T cells. Statistical calculation was performed using Two-way ANOVA. Statistical comparison with *p* > 0.05 is not shown, * = *p* ≤ 0.05, ** = *p* ≤ 0.01, *** = *p* ≤ 0.001, **** = *p* ≤ 0.0001.

**Figure 2 ijms-25-05895-f002:**
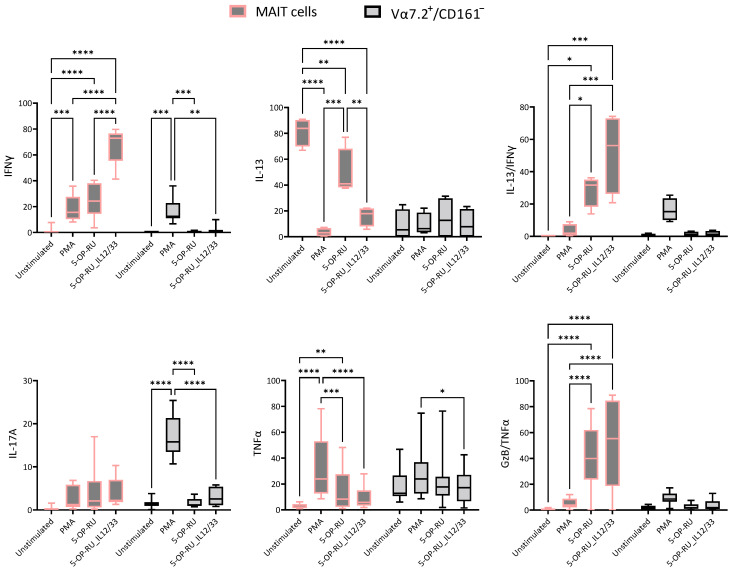
Intracellular cytokine-producing MAIT and Vα7.2+/CD161− T cells were compared between different stimulatory conditions. Grey color gradients with pink borders are MAIT cells and grey color gradients with black color borders represent Vα7.2+/CD161− T cells. X-axis represents different stimulatory conditions (Unstimulated, Phorbol myristate acetate (PMA)/ionomycin, 5-(2-oxopropylideneamino)-6-d-ribitylaminouracil (5-OP-RU), and 5-OP-RU/IL-12/IL-33) and Y-axis represents marker expression within as % of MAIT cells or % of Vα7.2+/CD161− T cells. Statistical calculation was performed using Two-way ANOVA. Statistical comparisons with *p* > 0.05 are not shown, * = *p* ≤ 0.05, ** = *p* ≤ 0.01, *** = *p* ≤ 0.001, **** = *p* ≤ 0.0001.

**Figure 3 ijms-25-05895-f003:**
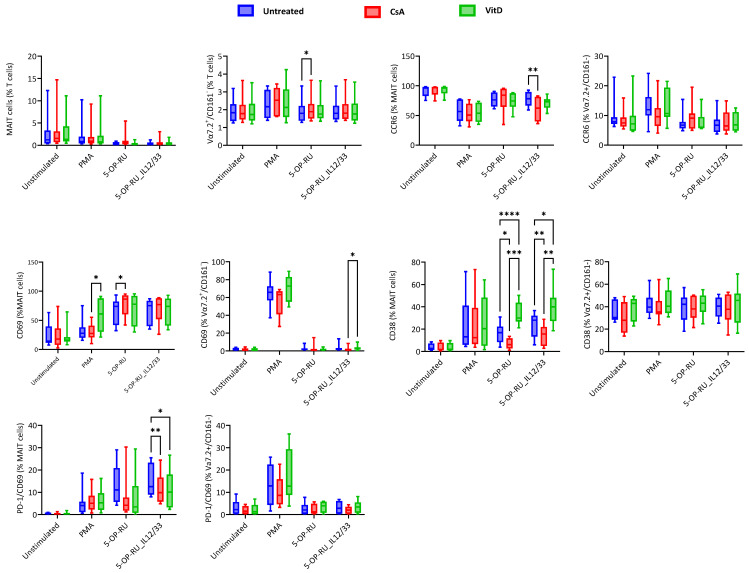
Treatment-based changes in cellular surface markers of MAIT and Vα7.2+/CD161− T cells under different stimulatory conditions. Blue color gradients represent untreated cells, red color gradients represent Cyclosporin A-treated cells, and green color gradients represent Vitamin D3-treated cells. X-axis represents different stimulatory conditions (Unstimulated, Phorbol myristate acetate (PMA)/ionomycin, 5-(2-oxopropylideneamino)-6-d-ribitylaminouracil (5-OP-RU), and 5-OP-RU/IL-12/IL-33) and Y-axis represents marker expression within as % of MAIT cells or % of Vα7.2+/CD161− T cells. Statistical calculation was performed using Two-way ANOVA to compare individual treatment groups (Untreated, Cyclosporin A, and Vitamin D3). Statistical comparisons with *p* > 0.05 are not shown, * = *p* ≤ 0.05, ** = *p* ≤ 0.01, *** = *p* ≤ 0.001, **** = *p* ≤ 0.0001.

**Figure 4 ijms-25-05895-f004:**
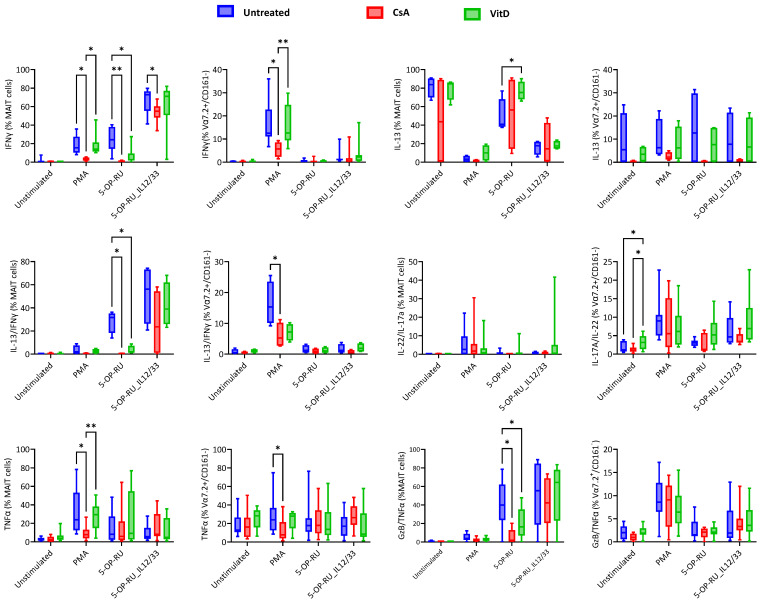
Treatment-based changes in intracellular cytokines of MAIT and Vα7.2+/CD161− T cells under different stimulatory conditions. Blue color gradients represent untreated cells, red color gradients represent Cyclosporin A-treated cells, and green color gradients represent Vitamin D3-treated cells. X-axis represents different stimulatory conditions (Unstimulated, Phorbol myristate acetate (PMA)/ionomycin, 5-(2-oxopropylideneamino)-6-d-ribitylaminouracil (5-OP-RU), and 5-OP-RU/IL-12/IL-33) and Y-axis represents marker expression within as % of MAIT cells or % of Vα7.2+/CD161− T cells. Statistical calculation was performed using Two-way ANOVA to compare individual treatment groups (Untreated, Cyclosporin A, and Vitamin D3). Statistical comparisons with *p* > 0.05 are not shown, * = *p* ≤ 0.05, ** = *p* ≤ 0.01.

**Figure 5 ijms-25-05895-f005:**
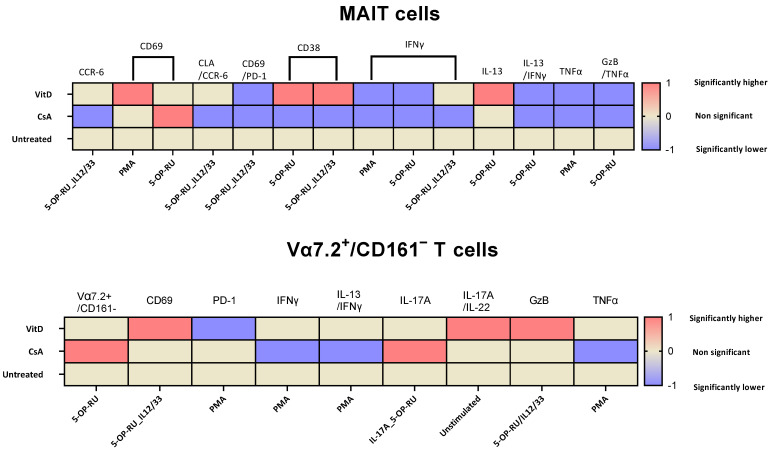
Summarizing the data found to be statistically significant between the treatment group comparison of MAIT and Vα7.2+/CD161− T cells. The heatmap gradient shows 0, which means nonsignificant (brown color), −1, which shows significantly lower (blue color), and 1, which indicates considerably higher (red color). On the top, the different markers are given; in the left corners, other treatments are mentioned (Untreated, Cyclosporin A, and Vitamin D3) and the stimulants (Phorbol myristate acetate (PMA)/ionomycin, 5-(2-oxopropylideneamino)-6-d-ribitylaminouracil (5-OP-RU), and 5-OP-RU/IL-12/IL-33) are given in the lower part of the table.

**Table 1 ijms-25-05895-t001:** Experiment designed to test the different stimulatory conditions and immunomodulatory drugs. Three different stimulatory conditions (PMA (Phorbol myristate acetate)/Ionomycin, 5-(2-oxopropylideneamino)-6-d-ribitylaminouracil (5-OP-RU), and 5-OP-RU/IL-12/IL-33) were used along with an unstimulated condition (control). All these conditions were incubated with or without immunomodulatory drugs (Cyclosporin A and Vitamin D3) for 24 h, and in the last 6 h, Brefeldin-A was added.

Tube No	Stimulation	Cyclosporin A	Vitamin D3
1	Unstimulated	−	−
2	PMA/Ionomycin	−	−
3	5-OP-RU	−	−
4	5-OP-RU+IL-12 and IL-33	−	−
5	Unstimulated	+	−
6	PMA/Ionomycin	+	−
7	5-OP-RU	+	−
8	5-OP-RU+IL-12 and IL-33	+	−
9	Unstimulated	−	+
10	PMA/Ionomycin	−	+
11	5-OP-RU	−	+
12	5-OP-RU+IL-12 and IL-33	−	+

**Table 2 ijms-25-05895-t002:** Monoclonal antibody panels for flow cytometric staining. A three-tube panel was used to perform cellular surface and intracellular staining after the immunomodulatory drug experiment. Tube 1 contains only cellular surface markers, and tubes 2 and 3 contain antibodies against surface and intracellular targets.

Tube	FITC	PE	PerCP/Cy5.5	PC7	APC	APC-H7	PB	PO
Tube 1	CLA	CCR6	Vα7.2	CD161	PD-1	CD3	CD69	CD45
Tube 2	cyIFN-g	cyIL-4/IL-13	Vα7.2	CD161	cyIL-22	CD3	cyIL-17A	CD45
Tube 3	Syto	cyTNF-α	Vα7.2	CD161	CD38	CD3	cyGzB	CD45

## Data Availability

The data supporting the study’s findings are available from the corresponding author upon request.

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
