# Peer review of "A Comprehensive Investigation of Stimulatory Agents on MAIT and Vα7.2+/CD161− T Cell Response and Effects of Immunomodulatory Drugs"

_ijms, 2024, doi:10.3390/ijms25115895_

Round 1

Reviewer 1 Report

Comments and Suggestions for Authors

This manuscript by Singh et al., reports the effects of two immunomodulatory drugs, Cyclosporin A (CsA) and Vitamin D3 (VitD) on in vitro cytokine functions by stimulated peripheral blood MAIT cells obtained from healthy donors. Flow cytometric analysis is used to assess surface markers and intracellular cytokine production. Overall, the manuscript is well written with appropriately justified experiments. For the most part, the results support the conclusions made. Below are the specific points the authors must address to improve the rigor of this research:

1. Although the authors state that "Not all MAIT cells identified with Vα7.2/CD161+ bind to the MR-1 tetramer; not all MR-1 binding tetramer express Vα7.2/CD161+" they do not explain why 5-OP-RU loaded MR-1 tetramers were not used to identify MAIT cells. Please provide the rationale for using Vα7.2/CD161 inseead of MR1 tetramers.

2. Please use CCR6 instead of CCR-6.

3. Lines 73-75: Provide references.

4. What is the rationale for comparing MAIT cells with Va7.2+/CD161-neg cells? How are they distinct from other non-MAIT T cells?

5. Line 107: "Untreated data were used to calculate the significant difference between different stimulants to avoid treatment biases", please clarify...was untreated data used for background subtraction?

6. Lines 119-121 and Figure 1: How is CD69 not significant for PMA stim? Is it because most activated cells are missed by the flow gating strategy for MAIT cells? Even with Can the auhors comment if it is because of CD161 downregulation or AICD?

7. IL-13 gating: It is highly unlikely to have that high frequency of IL-13 expression on unstimulated MAIT cells from healthy donors. The supplementary figures showing flow plots should have axis labels and scale. The IL-13 gate is supposed to be shifted to the right based on unstimulated plot and the axis scale will give an idea where it lies relative to 0 on scale. To confirm the gating, same gates must be applied to PMA-stimulated CD4 T cells that are more likely to show a clear positive and negative population of IL-13-producing cells.

8. Line 181 and elsewhere: "However, 5-OP-RU-stimulated, and CsA-treated Vα7.2+/CD161- T cells showed significantly higher numbers than the untreated cells". This is really frequency not numbers.

9. Figures: please remove markings for non-significant comparisons and only show significant differences. It is distracting to see "ns" everywhere and would be easy to spot significant differences.

10. Lines 303-305: Is there any references for high IL-13 production by MAIT cells? Please cite them here.

Comments on the Quality of English Language

Minor edits needed. Example: Use untreated instead of "untreat" and correct line 164.

Reviewer 2 Report

Comments and Suggestions for Authors

Comments on the Quality of English Language

Round 2

Reviewer 2 Report

Comments and Suggestions for Authors

the authors have improved the quality of the paper